# Relationship Between Basic Human Values and Decision-Making Styles in Adolescents

**DOI:** 10.3390/ijerph17228315

**Published:** 2020-11-10

**Authors:** Javier Páez Gallego, Ángel De-Juanas Oliva, Francisco Javier García-Castilla, Álvaro Muelas

**Affiliations:** 1Department of Psychology, Faculty of Biomedical Sciences and Health, Universidad Europea de Madrid, Villaviciosa de Odón, 28670 Madrid, Spain; 2Department of Theory of Education and Social Pedagogy, Education Faculty, UNED, 28040 Madrid, Spain; adejuanas@edu.uned.es; 3Department of Social Work, Law Faculty, UNED, 28040 Madrid, Spain; fjgarcia@der.uned.es; 4Department of Education, Universidad Villanueva, 28034 Madrid, Spain; alvaro_muelas@hotmail.com

**Keywords:** social values, decisión-making, adolescents, moral development, social behavior

## Abstract

This paper examines the relationship between decision-making styles and values of Spanish adolescents and analyses the role of age and gender on their use of *adaptive* and *maladaptive* decision-making styles. The scientific literature suggests that decision-making consists of different stages through which individuals reach a solution to their dilemmas. An ex post facto quantitative, non-experimental research design was used and applied to a sample of adolescents of Madrid (España). The Flinders Adolescents Decision-Making Questionnaire (FADMQ) by Mann as well as the Schwartz Values Scale (SVS) were also used. Correlation analysis was used to analyze the decision-making styles and values of adolescents using the variables gender and age to classify the sample. The study concludes that adolescents who use an *adaptive decision-making style* tend to pursue mastery of the values *Self-direction, Stimulation, Achievement*, and *Power*, whereas adolescents who use a *maladaptive style* tend to shy away from the value *Self-direction* and are more conservative. In terms of gender, the results for both females and males coincide in the significant correlations found between their decision-making styles and values. In terms of age, the correlations between values and decision-making styles are higher and numerous in younger adolescents. We conclude that the relationships verified could help educators to engage and act on the development of specific decision-making training programs based on the values of adolescents.

## 1. Introduction

Although not all individuals share the same convictions, values play an important role in our society and help to identify us via emotional, cognitive, and behavioral aspects of our personalities. As such, there is an ongoing major debate in the literature about the relationship between values and decision-making. However, everything seems to indicate that values do influence decision-making and act as criteria for judging and evaluating what is deemed to be desirable, which has a direct impact on the behavior of adolescents [1]. Nevertheless, when approaching this subject, it must be taken into account that values change from person to person and even as people age due to the effect produced by beliefs, customs, social interactions, and cultural and historical contexts [2].

In relation to this, in recent decades, interest in the Schwartz Theory of Basic Human Values [3] has grown in the scientific community. This model is based on a set of largely accepted universal values and aims to measure interests and incentives while highlighting the importance of motivational content in forming values. In this way, Schwartz, op. cit., defines values as a cognitive construct that is used to evaluate which actions and cognitions are desirable and which are not. In this sense, Schwartz (op. cit.) analyzes in the development of his theoretical model what values are present in all cultures. Thus, the ten values of the model have a universal character. Schwartz [4] argues that social values are fundamentally the result of needs arising from three universal requirements of the human condition. First, the needs of *the individual as an entity* that develops within the social and natural world. Second, the need to establish *relationships between individuals and groups*. Third, the need for *responsible social conduct*, which aims to encourage respect in individuals toward the welfare and coordination of groups. From these three basic requirements, Schwartz, op. cit., derived ten types of fundamental or universal values common to all cultures, which present both congruent and conflictive relationships. These values are arranged in a circular continuum that places those with congruent attainments and achievements closer together than those that differ from one another (Figure 1).

This structure reflects a motivational continuum that is organized in four higher values that are located opposite to each other to form a “two by two” relationship of pairs of bipolar dimensions. The first pair, *Openness to change* versus *Conservation*, groups and combines the factors of *Stimulation* and *Self-direction* in contrast to the values of *Safety*, *Conformity,* and *Tradition*. These values classify people’s impulses in terms of following their own emotional and intellectual interests as opposed to those that encourage compliance and self-restriction. The second pair, *Self-enhancement* versus *Self-transcendence*, groups and organizes the values of *Power* and *Achievement,* which focus on self-interest, in contrast to *Universalism* and *Benevolence*, which focus on concern for the wellbeing of others. Given that the values proposed by Schwartz correspond to intrinsic needs, it is important to determine the motivational content that constitutes each individual one. In his research, Schwartz, op. cit., collected samples from more than 40 culturally diverse countries and found that the meaning of the values in most of the groups was quite similar, thus obtaining empirical support for his theoretical model [5].

In the context of our research, another important source of knowledge is the Schwartz Values Survey (SVS) in order to analyze the predominant values structure in adolescents. At the beginning of the twenty-first century, several studies were performed that used Schwartz’s model to study young people’s values in relation to drug consumption, criminal behavior, and parenting styles [6]. The relationships adolescents establish with their peers [7] and values structure in which the preferred values of 505 Spanish students between the ages of 15 and 19 were analyzed [8].

The foremost question, when it comes to adolescents, is to determine the most important values that guide their actions. In fact, as part of the decision-making process, the influence of the context demands on young people’s lifestyles means that some behaviors prevail over others, although they might not always be positive for the individual or society [9]. Phenomena such as immediacy, dissatisfaction, incorrect attributions, and false expectations lead many adolescents to make hasty decisions in order to solve their problems and avoid stressful situations and possible communication barriers. Yet, thanks to their values, adolescents are able to guide decision-making in relation to their feelings, behavior, and intentions. The terminology associated with the decision-making process is broad. To put it simply, decision-making is a conflict between many alternative solutions, each with their own diverse implicit values [10]. More recently, Hastie and Dawes [11] have gone even further and conceptualized decision-making as the response to circumstances based on the availability of alternative actions, the decision maker’s expectations of the probability of each one of the alternatives occurring, and the associated consequences. In addition, the consequences of each alternative have to be hierarchized in a continuum formed by individual goals and personal values. This last conceptualization perfectly summarizes the process whereby a subject is faced with a decision-making situation. This has been the object of study in several disciplines [11,12,13,14,15,16,17].

Until recently, research on adolescent decision-making was based on the normative model of decision making by Baron and Brown [18]. This theory focuses on five steps that aim to identify the decision-making approach which involves: establishing alternative solutions, identifying the evaluation criteria of the alternatives, assessing the consequences associated with each alternative, gathering additional information and seeking help from third parties, and finally making decision-making executing the alternative [19].

Supported by these theoretical assumptions, there is a large and growing scientific literature that has attempted to examine intuitive adjustment in adolescent decision-making. These studies have confirmed that, without any specific training, adolescents intuitively learn to create a list of solution alternatives [20,21,22,23,24,25]. However, the attempt to establish standard compliance with the stages that comprise the decision-making process of adolescents reveals unequal development according to different age groups. As a result, adolescents between the ages of 12 and 14 are likely to find it difficult to identify decision-making situations and to put decisions into practice. In turn, adolescents between 15 and 16 may be unable to seek additional information and require social support to make decisions. Finally, many older adolescents seek less help from others as peer pressure decreases [26,27]. In fact, studies by Haller, Bang, Bahrami, and Lau [28] highlight that younger adolescents (12–13-year-olds) were less competent in identifying evaluation criteria for different alternatives in decision-making processes than older adolescents (17–18-year-olds). This finding has huge implications in terms of values content, especially considering that immediacy is pathognomonic of the adolescent stage. This is why the identification and enumeration of the consequences associated with previously identified alternatives constitute the most difficult task for adolescents to perform.

Taking into account the variations in the decision-making process and the impact of values when choosing alternatives, studies on decision-making coping patterns are based on the conflict theory by Mann, Harmoni, and Power [29]. According to this model, in decision-making, there is a discrepancy between the alternative options and/or their implicit values. Grootens-Wiegers, Hein, Van den Broek and de Vries [30] used the classification of four decision-making coping patterns differentiating between vigilance, complacency, hypervigilance, and avoidance. Subsequently, Mann, op. cit. grouped these four patterns into two fundamental clusters: On the one hand, the *adaptive* or *vigilant* style and, on the other, the *maladaptive* style comprising *complacency*, *hypervigilance*, and *avoidance*. Thus, the *adaptive style* corresponds to the careful observation of the elements that comprise deciding and reflecting on alternative solutions and extracting the pros and cons of each before committing to a final decision. In contrast, the *maladaptive style* involves the taking of decisions that seek to satisfy other people’s objectives, hastily contrived solutions, uncertainty which leads to feelings of panic that hinder careful consideration of alternatives, and/or delegating decision-making to a third person.

This study considers the fact that decision-making is closely related to values which, in turn, are closely related to the behavior and life goals of adolescents. Therefore, it is necessary to address the understanding of the interaction of these processes in a time of change as decisive in people as adolescence. In this sense, the main contribution of this study is the analysis of the relationship between decision-making processes and the values of adolescents. Simultaneously, we intend to examine the state of adaptive and non-adaptive decision-making styles in adolescents using the variables of age and gender.

The hypotheses derived from these objectives are:Adolescents who identify with values of Self-transcendence, Openness to change, and Self-promotion employ more adaptive decision-making strategiesAdolescents who identify with Conservation values, employ more maladaptive decision-making strategies

On the other hand, hypothesizing the use of decision-making strategies based on sex, the hypotheses are as follows:Women who identify with values of Self-transcendence, Openness to change, and Self-promotion employ more adaptive decision-making strategiesMen who are identified with values of Self-transcendence, Openness to change, and Self-promotion, employ more adaptive decision-making strategies

Regarding the analysis based on age, the hypotheses are as follows:Older adolescents who are identified with values of Self-transcendence, Openness to change, and Self-promotion employ more adaptive decision-making strategiesOlder adolescents who identify with Conservation values employ more adaptive decision-making strategies

## 2. Materials and Methods

The study is based on a quantitative, descriptive, non-experimental, ex post facto research design. In turn, in accordance with previous findings and the study objectives, we also considered the possible relationship between the predominant decision-making style and values profile of Spanish adolescents. We also analyzed whether this possible relationship is influenced by the variables age and gender.

By the same token, we considered whether there are differences in the use of *adaptive* and *maladaptive decision-making styles* by adolescents based on the aforementioned age and gender variables.

### 2.1. Participants

The study involved 385 students from nine compulsory secondary education and baccalaureate centers in central Spain (males, *n* = 193, 50.13%; females, *n* = 192, 49.87%). The initial sample was 2112 adolescents of the Community of Madrid. Subsequently, the sample size was calculated, yielding a result of 385 subjects, with a confidence level of 95% and a margin of error of 5%. With this data, a random resampling was carried out on the initial sample through proportional allocation according to strata to which the region to which the students belong, the type of study, the type of center—public, private, or subsidized was taken into account—and gender.

The participants were aged between 13 and 19 (M ± DE: age: 15.55 ± 1.618).

### 2.2. Materials

Two measuring instruments and an initial survey were used to collect sociodemographic data. The first of these, the Flinders Adolescents Decision-Making Questionnaire (FADMQ) by Mann, op. cit. and reviewed by Colakkadioglu and Deniz [31], covers the dependent variables: the *adaptive style* and the *maladaptive style*, involving those strategies typical of complacency, hypervigilance, and avoidance.

The FADMQ scale presents the following psychometric properties.

Regarding validity, the confirmatory factor analysis shows the existence of two factors that explain 40% of the variance. The first factor includes Hypervigilance, Avoidance, and Complacency styles. This factor explains a 25.27% variance and has an eigenvalue of 7.33 (explained variance = 25.27%; eigenvalue = 7.33).

On the other hand, the second factor groups together the items that evaluate the adaptive decision-making style.

Regarding reliability and internal consistency, each of the two subscales-adaptive styles and maladaptive decision-making style are analyzed separately. Thus, the internal consistency of the “maladaptive” factor is 74 (α = 74). On the other hand, the internal consistency of the “adaptive” factor is 78 (α = 78)

In turn, the SVS scale, by Schwartz [3] and adapted by Páez and De-Juanas [32] for the Spanish adolescent population, measures the participant’s preferred values using Schwartz’s Theory of Basic Human values. This measures the independent variables of the preferred values and includes the discrete ordinal variables: *Self-direction*–independent thought and decision-making, with a tendency for creativity, freedom, choosing one’s own goals, curiosity, and independence; *Stimulation*–a characteristic of this value is self-affirmation, which is linked to the search for stimulation and life challenges, a varied lifestyle, and daring; *Hedonism*–involves the individual pursuit of pleasure through living positive emotional experiences; *Security*–the desire for stability, both in the social and personal environment. As a result, attaining macro, meso, and microsocial stability is prioritized, and positive interpersonal relationships are assured via a commonly accepted social order. This value embodies the aspiration for individual and group survival; *Conformity*–avoiding harm to other individuals through the fulfillment of social norms that protect freedom and respect for others. The main attributes are obedience, self-discipline, and good manners; *Tradition*–respect for the customs and ideas imposed by traditional norms and culture. This reflects attitudes of respect for tradition, devotion, humility, and moderation; *Achievement*–refers to the search for social approval and recognition according to social standards. It embodies values such as: ambition, ability, influence, and intelligence; *Power*–the prioritizing of control and dominance over other people and/or groups that belong to the same field; *Benevolence*–alludes to preserving positive relations in order to preserve the wellbeing of individuals and groups and is linked to values such as help, honesty, forgiveness, loyalty, and responsibility; and lastly, *Universalism*–the individuals described by this value seeks to ensure the survival of individuals and groups when the resources on which life depends are under threat. It integrates attitudes such as personal maturity, tolerance, wisdom, environmental protection, general social welfare, social justice, equality, etc.

The SVS scale presents the following psychometric properties.

In the first place, with respect to validity, 10 factors were obtained whose content coincided with the description of the values of the Schwartz model. The fit indices of the factor analysis showed results that confirmed the adequacy of the factorial model to the theory. The Chi square obtained a value of 3.15 (χ^2^/gL = 3.15) (χ^2^/813 = 2561.5, *p* < 0.01), the GFI coefficient had a value of 0.92 (GFI = 0.92), the CFI was 0.90 (CFI = 0.90), the result of the NNFI coefficient was 0.90 (NNFI = 0.90) and, finally, the RMSEA value was 0.07 (RMSEA = 0.07). These results support the hypothesized theoretical model.

Second, regarding the total reliability of the instrument, measured with the Cronbach’s alpha coefficient, it reached a value of 0.94.

### 2.3. Procedure

The collective questionnaire was presented in a *pencil and paper* format, in a single session, in all education centers during school hours. The participants were informed of the purpose of the study, and participation was voluntary, anonymous, and under instruction. Similarly, parents and guardians were informed about the purpose of the study and signed a voluntary consent form that allowed their children to participate. This study adhered to the guidelines found in the Declaration of Helsinki [33], which establishes ethical principles for research involving human subjects. The Research and Doctorate Commission of the Faculty of Education carried out the evaluation of the ethical principles of the research, granting its approval as stated in the corresponding certification.In addition, during and after the entire research process, we proceeded in accordance with Spanish Organic Law 15/1999 of 13 December on the protection of personal data.

## 3. Results

First of all, the present results are part of a larger investigation.

### 3.1. Preferred Values in Adolescents

Table 1 shows the data from the responses to the questionnaire. The average score for each value category for the study sample is presented in order of highest to lowest. As a result, the highest to lowest preference for each of the ten value categories that constitute the model can be clearly seen.

In Table 1, the preferred values given the highest scores by the study sample are *Benevolence, Self-direction, Hedonism*, which are grouped in the Schwartz model, and *Conformity*, which is close to *Benevolence* on the continuum. All of them have an average score of >7.

In contrast, the values with the lowest preference index are *Stimulation* and *Power*, both with an average of <6.

All values show what can be interpreted as a small dispersion that oscillates between s = 0.90 and s = 1.41.

### 3.2. Use of the Adaptive Decision-Making Style

The scores obtained by the study sample on Mann’s decision-making style scale (1988) show an average of >10 ($\underline{x}$ = 12.02). Considering that the scores on this scale range from 0 to 21, this average statistic places the sample slightly above the medium score in the range.

In addition, the standard deviation shows a moderate dispersion (s= 2.721), even though the maximum and minimum scores indicate the existence of subjects whose scores were at both extremes (Min = 3; Max = 21).

### 3.3. Use of the Maladaptive Decision-Making Style

The average score on this subscale is close to 4 ($\underline{x}$ = 3.87). However, the range on this subscale is lower than the *adaptive style* subscale (ranging from 0 to 15 points) and the mean is well below the average score in the range (7–8 points).

The maximum (Max. = 14) and minimum (Min. = 0) scores obtained by the study sample in this subscale verify that there are subjects whose scores were at both extremes of the scale. However, the standard deviation (s = 2.598) shows little dispersion of the scores with respect to the mean. As a result, a large number of subjects obtained low scores which indicates that a slight majority use *maladaptive* strategies.

### 3.4. Relationship between the Predominant Decision-Making Style and Values Profile of Adolescents

Facing the decision-making process involves putting into practice a series of strategies on the spectrum between *adaptive* and *maladaptive styles*. The comparatively frequent use of either of these styles defines decision-making coping patterns. Thus, taking the contrast in the relationship between the decision-making style and the ten values of Schwartz’s model as an approach, the results of the correlation study can be seen in Table 2.

The results show a significant correlation between the *adaptive* decision-making style and the values *Self-direction* (*r* = 126; *p* = 0.014) and *Stimulation* (*r* = 110; *p* = 0.032), with a probability of less than 05, and also *Achievement* (*r* = 207; *p* = 0.000) and *Power* (*r* = 140; *p* = 0.006), with a probability of less than 01. These values belong to the higher categories *Self-enhancement* and *Openness to change*. In contrast, there was only a significant correlation between *maladaptive* strategies and the value *Self-direction* (*r* = −0.128; *p* = 0.013). However, this negative value indicates that subjects with a high score in the *Self-direction* category use limited *maladaptive* decision-making strategies, whereas a low score would indicate the opposite.

### 3.5. Relationship between the Predominant Decision-Making Style and Values Profile Based on Gender

The classification of the study sample into groups comprising the different values of the variables age and gender allows for a more detailed analysis of the use of decision-making strategies and their relationship to the structure of the values. Thus, the correlation between values and decision-making styles based on gender shows a high incidence in the female group in the case of the *adaptive style*. The five values which obtained a significant correlation were: *Self-direction* (*r* = 146; *p* = 0.045), *Benevolence* (*r* = 158; *p* = 0.029), *Conformity* (*r* = 182; *p* = 0.012), *Achievement* (*r* = 205; *p* = 0.004), and *Tradition* (*r* = 196; *p* = 0.006). There were no significant correlations between the *maladaptive style* and the values (see Table 3).

The positive correlations between values and the *adaptive style* in the female group belong indistinctly to the four higher category values. However, *Benevolence*, *Tradition,* and *Conformity* are closely related, which enables us to define a values style characterized by the pursuit of the common good and respect for socially established norms and agreements. The male group obtained a positive correlation between the *adaptive decision-making style* and the value *Achievement* (*r* = 203; *p* = 0.005) and a negative correlation between the *maladaptive style* and the value *Self-direction* (*r* = −202; *p* = 0.005). Both significant correlations in the male group coincide with significant correlations found in the female group and in the analysis of the sample as a whole.

### 3.6. Relationship between Predominant Decision-Making Style and Values Profile Based on Age

The analysis of the relationship between the variables under study according to age groups reveals very different profiles depending on whether the *adaptive* or the *maladaptive style* is analyzed. In the case of the *adaptive decision-making style* and values, the younger age groups produce more correlations than the older age groups (Table 4). Consequently, the 13-year-old group obtained a significant correlation between the *adaptive style* and the value *Conformity* (*r* = 290; *p* = 0.018). The 14-year-old group obtained five significant correlations between the *adaptive decision-making style* and the values *Self-Direction* (*r* = 264; *p* = 0.038), *Stimulation* (*r* = 314; *p* = 0.013), *Hedonism* (*r* = 268; *p* = 0.035), *Achievement* (*r* = 323; *p* = 0.011), and *Power* (*r* = 419; *p* = 0.001). Whereas, the 18-year old group obtained two significant correlations with the values *Achievement* (*r* = 431; *p* = 0.008) and *Power* (*r* = 531; *p* = 0.001), the probability was less than 01 in both cases.

In contrast, in the *maladaptive style,* the highest number of significant correlations were obtained by the older age groups at the intersection of values (Table 4).

The 15-year old group obtained a significant correlation between the *maladaptive style* and the value *Power* (r = 384; p = 0.001). The 18-year old group obtained up to three significant correlations between the *maladaptive style* and the value *Achievement* (*r* = −0.328; *p* = 0.047). However, it should be noted that all the correlations were negative. Lastly, the 19-year old group obtained a significant correlation between the variables and the value *Power* (*r* = 579; *p* = 0.038).

## 4. Discussion

Values have a very important role in the study of human behavior due to their explanatory role in motivating behaviors. That is, knowing the value system of people allows inferring attitudes and anticipating behavior.

This predictive character of the axioms is reflected in the model of universal values of Schwartz [3]. In this sense, as this theory proposes, values arise from three universal motivations. In addition, Schwartz (op. cit.) offers a description of the behavioral development of these values. Thus, if the expected behaviors of each value are analyzed, it is seen that the Self-direction, Stimulation, Achievement, and Power values, which obtain significant correlations with the adaptive decision-making style for the global group and for classifications based on sex and age, they have behavioral developments closely related to the decision-making process.

In this sense, according to Schwartz, the Self-Direction value is closely related to decision-making and the choice of one’s own goals.

In the same way, the stimulation value implies the search for novelties and varied life, so these people take more risks in decision-making.

Similarly, the Achievement and Power values imply a constant search for resources as well as prestige and power over others. To do this, people who identify with these values have to make frequent decisions due to the highly competitive nature of these values.

In other words, identification with one or more of these values implies the frequent performance of the decision-making task and, therefore, these people are more likely to develop adaptive decision-making strategies. Thus, the results are consistent with the definition of universal values proposed by the author of the theoretical model.

These results are more clearly shown in the group of women when the relationships based on sex are analyzed. Thus, this greater number of significant relationships in the group of women for the values Self-direction, Benevolence, Conformity, Achievement, and Tradition can be explained by what Guilligan [34] defines as female ethics, according to which, the moral development of women is based on the principle of responsibility, while that of men is based on the principle of justice. However, this idea is refuted by Kolhberg, who proposes a similar model of moral development for both sexes.

On the other hand, the non-existence of differences in the preferential use of decision-making strategies based on age would be explained by the adolescents’ lack of prior practice in this competence. In this way, it is not until the final years of adolescence that adolescents have to face the first vocational choices and, at the same time, the first autonomous decisions [35].

All in all, the results obtained in this research show the close relationship between values and decision-making. Therefore, knowing what a person’s value system is would allow us to infer their decision-making capacity and vice versa. This means that training in decision-making competence also involves questioning and defining the value system.

The wide acceptance of Schwartz’s Theory of Basic Human Values has propitiated its use in the study of values. However, given that previous evaluations on preferred values in adolescents have been performed in relation to other contrast variables, we have been unable to find studies that establish a hierarchical list of young people’s preferred values with which to compare the results of this study. However, some studies have approached the field in question using methods that allow us to contrast the results obtained. One of the most important studies was developed by Depaula, Piñeyro, Clotet, and Nistal [36], in which an objective very similar to that of this study was proposed, by trying to relate different levels of cultural intelligence, leadership styles, and human values to the predominant decision-making style. However, the approach used to measure the decision-making style–Operational Decision Making Scale (ODMS) adapted from the original Tacit Knowledge Survey [37]–uses two categories to define styles: intuitive and analytical, which differ from the *adaptive* and the *maladaptive style* categories of this study.

The definition of both styles enables parallels to be drawn between both categorizations. Thus, the intuitive decision-making style is defined as the analysis that is performed out of habit to facilitate a quick response once the main elements of the situation have been identified; it is very useful in highly complex contexts. In contrast, the analytical decision-making style is characterized by the use of cognitive with rational principles of analysis. Although the definition of both styles does not correspond to that given to the *adaptive* and *maladaptive styles*, there are features that liken them to each other. As a result, the spontaneous and habit-like nature of the quick responses characteristic of the intuitive style resembles the *maladaptive style*, whereas the detailed study of alternatives by means of pre-established criteria is characteristic of the *adaptive style*. However, these comparisons must be interpreted with caution, given that the theoretical definition of the intuitive and analytical styles does not include a negative component per se. The optimum use of each depends on the context or the situation in which they are used. This is not the case with the categories used in this study, which views the preferential use of *maladaptive* decision-making strategies negatively.

Thus, the data obtained in this study are in contrast to those obtained by Depaula, Piñeyro, Clotet, and Nistal, op. cit. They found that the subjects who defined themselves by the three values in the higher-order category—*Openness to change*–*Self-direction*, *Hedonism*, and *Stimulation*–showed a preference for independent attitudes and actions, and the pursuit of new experiences, which gives priority to the intuitive style (*p* = 0.03). They also show an almost significant relationship with the value *Achievement* (*p* = 0.071), giving priority to decisions linked to obtaining personal benefit through rational decision-making processes. These authors hypothesize that the difference between *Hedonism* and the other two values associated with *Openness to change* may be due to the fact that the pursuit of pleasure and gratification during the decision-making process help to counter the stress of uncertainty. However, the results found in the relational analysis of this study show the opposite result, given that there is a significant correlation between the values *Self-direction* and *Stimulation,* and *Achievement* and *Power* and the preferred use of the *adaptive style*, which contrasts with the decision-making strategies described in the intuitive style. After analyzing the results in the individual groups comprising the sociodemographic variables, similar conclusions were also found in the total of the final sample group. Thus, there is a significant correlation between the three values of *Openness to change* and the two values of *Self-enhancement* with the *adaptive decision-making style* in the female group, in the 13–14-year old group, and in those who study the 2nd and 4th years of compulsory secondary education.

In contrast, the similarities found in the 18-year old group are noteworthy. In this case, there is a significant correlation between the values *Self-direction, Hedonism,* and *Achievement* and *maladaptive* strategies. These results are almost identical to those described by Depaula, Piñeyro, Clotet, and Nistal, op. cit.

Despite the differences and similarities with the study by Depaula et al., op. cit., the statistically significant relationships found in both cases confirm the relevance of values in the decision-making process. It can be concluded that values are a conditioning factor in the cognitive processes used to face new experiential situations. Similar conclusions can be found in previous studies [38] in which a significant relationship is established between the use of the intuitive style and the preference for the values *Self-direction* and *Stimulation*, substantiating that values are important in the decision-making process under situations of risk.

On the other hand, regarding the values that do not have a significant relationship with the two decision-making styles, it is important to observe that each of the universal values proposed by Schwartz in his theoretical model has a motivational and behavioral development [4]. That is, they arise or respond to a specific need, and their satisfaction is carried out through specific behavioral or cognitive actions.

In this sense, some values respond to needs more related to decision-making than other values. In this way, the behaviors that are carried out in coherence with this value suppose a greater performance of the decision-making task.

In this way, if taken as a reference the definition that Schwartz [3] makes of each value, the axioms that break with tradition and that promote greater change (*Self-Direction* and *Benevolence*) have more decision-making behaviors. Something similar happens with the Power and Achievement values, to which important decision-making is attributed to achieve the ambitious objectives proposed by the person who identifies with them.

However, there is a wide group of values that have little decisive content in a motivational foundation and in behavioral development, either because of their conservative nature or because, simply, their motivation does not incline people who identify with them to take numerous or important decisions.

For this reason, the *Stimulation*, *Hedonism*, *Power*, *Security,* and *Universalism* values do not receive any significant correlation.

Consequently, the values that obtain the highest number of significant correlations, as well as the axioms that do not receive any, allow the design of axiological profiles based on their performance of the decision-making task [39].

This conclusion is fundamental when designing training actions for decision-making since the prior identification of the axiological profile of the people would allow a greater and better adjustment of the objectives to be established and the concrete actions to be taken.

## 5. Conclusions

The findings in this study indicate that most participants demonstrate a values profile defined by the prosocial value *Benevolence* and more individualistic values such as *Self-direction* and *Hedonism,* as well as a predominantly *adaptive decision-making style*. The fundamental objective of this study was to determine the values and preferred values of adolescents. Evidently, determining their decision-making style was also a priority. This was especially important within the context of obtaining a clear overview of the behavior of both variables due to the educational implications that could lead to formative actions focused on helping adolescents make responsible and more accurate decisions. Providing, of course, that the best decisions adolescents can make are those that are intimately related to values regarding everything they profess to want and pursue. In light of the results, we can conclude that there is a relationship between the variables’ *decision-making style* and *values*. However, there was a positive correlation between the *adaptive decision-making style* and the values *Self-direction, Stimulation, Achievement,* and *Power,* while the only significant and negative correlation was between the *maladaptive decision-making style* and the value *Self-direction*.

A tentative interpretation of the significant correlations found in the *adaptive style* indicates that values belong to a profile defined by the pursuit of stimulating situations, the attainment of objectives for personal benefit, the ability to cope with low-risk decision-making situations, and the desire to command and have power over others. Subjects with this profile assume consequences more easily and are less resistant to change than subjects at the opposite pole of Schwartz’s model (Figure 1). This can generate a less stressful situation about the future and a greater mastery of the circumstances, which allow the subject to adopt *adaptive* decision-making strategies. This fact is corroborated by the significant correlation between the *maladaptive style* and the value *Self-direction*. This correlation is negative, which means that the subjects who define their values by this value are those who least employ the *maladaptive* decision-making strategies. In other words, adolescents who are more capable of generating alternatives and maintain independent opinions are less prone to use *maladaptive* strategies. The results point to *Self-direction* as a determining variable in the use of one or other of the decision-making strategies. This highlights the importance of making adolescents aware of their own values and teaching them to take into account the values and interests of others, which may come into conflict with their own.

In turn, if we consider the relationship between the predominant decision-making style and the values profile of adolescents according to gender, we find that there are differences in each group. However, the differences in the male group are very small, while for the female group they are more significant, though still modest. The limited correlations found in the male group—*Achievement*—do not define a conclusive profile. However, correlations with the values *Self-direction*, *Benevolence*, *Conformity*, *Achievement*, and *Tradition* in the female group highlight a profile concerned with the social dimension of actions, the pursuit of the common good, and respect for the norms and customs of society and culture. It also highlights an interest in achieving individual objectives of *Self-enhancement*, characteristic of the value *Achievement*. In addition, given that this value is the exact opposite of *Benevolence*, we could consider the existence of two differentiated values profiles in the female group that employ *adaptive* decision-making strategies. The results may lead us to consider that values cultivated, fundamentally, within the family and at school differ according to gender. This is perhaps due to strong historical and cultural roots, something that other studies have already highlighted.

Moreover, the study on the predominant decision-making style and values profile of adolescents according to age yield inconsistent results. This analysis was of particular interest to this study because society frequently justifies and attributes certain adolescent behavior to immaturity and lack of experience, even though the results do not support significant findings in this regard. However, significant correlations are found in the 13, 14, and 18-year old age groups for both styles. The results represent a grouping of correlations in the younger age groups, albeit with the isolated case of the 18-year old group. However, the values profiles described by the significant relationships in the use of *adaptive* strategies are different in each group. Consequently, in the 13-year-old group, the values *Self-direction* and *Conformity* are given priority. These values are not in conflict, neither are they congruent, therefore, we might expect to see two differentiated values profiles highlighted in the use of *adaptive* decision-making strategies. Something similar occurs with the 14-year-old group. The significant correlation with the values *Self-direction* and *Power* does not allow for the definition of a single values profile, which leads us to conclude that there are two differentiated profiles that are more prominent in the use of *adaptive* strategies. In addition, in the 18-year old group, there are two significant correlations between the variable *adaptive decision-making* style and the values *Power* and *Achievement*. In this case, both values are congruent as they are represented together in Schwartz’s universal values model, so a single values profile can be defined that is more prominent in the use of *adaptive* style decision-making strategies.

For all this, the results found in this research invite to design educational strategies aimed at training adolescents in the correct use of decision-making strategies.

In this sense, as has been observed, identification with openness values implies greater use of adaptive strategies. This trend is related to the individual safety of the people who make the decisions. Furthermore, other previous studies [3] have verified the relationship of the psychological well-being variables with the use of adaptive decision-making strategies.

Therefore, below are some guidelines for the design of training actions in decision-making:-The results show the relationship between openness and achievement axiological profiles with the use of adaptive decision-making strategies. Therefore, any training strategy that aims at training in decision-making must go through knowing the axiological profile of the learners to adjust the content of the same.-In the same way, when finding relationships between values and decision-making styles based on sex, training strategies in decision-making should be different. In this sense, actions to develop decision-making competence should be broadly intended for all men except for those who identify with the Achievement value. Rather, these same actions should be directed only at women identified with axioms related to personal self-promotion and tradition.-Training in decision-making competence must be approached holistically by all educational agents. Thus, formal education settings must offer specific training in rational decision-making methodologies. Along these lines, participatory teaching methodologies are a very important way to introduce this type of content in the classroom. On the other hand, non-formal education projects must offer activities that involve an experiential practice of the decision-making task. Finally, informal education contexts, such as the family or peers, should favor the practice of this competence in real situations.-Finally, training in adaptive decision-making strategies should be offered from early childhood, so that people gradually integrate a rational criterion in the evaluation of situations and alternative solutions. In this way, by reaching neurological maturity to make decisions rationally and autonomously (around the age of 15), people will be able to fully perform this competence.

Finally, regarding the limitations and future research, the data presented in this document may constitute a unique reference framework for the sample studied. However, although the sample size is considerable, it should be noted that conducting a resampling among a group of adolescents in Madrid (Spain) may offer a specific view of the results that limits their generalization to other populations. Having said this, we consider it important to replicate the study in other populations. Moreover, we believe that it is important to explore basic human values and decision-making styles throughout their lives and not only in a cross-sectional study. Furthermore, we are aware that the objective we set ourselves in this study did not contemplate the use of other measures that could be used to conduct other analyses (for example: psychological well-being, autonomy, etc.). We believe that this may be of great interest for future research.

## Figures and Tables

**Figure 1 ijerph-17-08315-f001:**
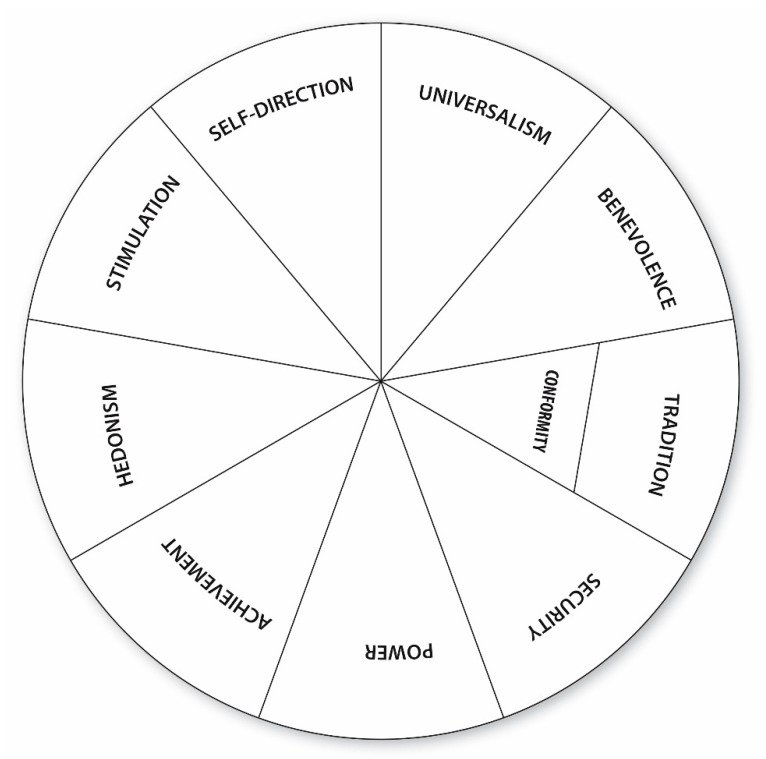
Theoretical model of relations among ten types of values.

**Table 1 ijerph-17-08315-t001:** Preferred values.

	N	M	SD	MAX	MIN	α
Benevolence	385	7.296	0.9666	45	5	0.76
Self-Direction	385	7.282	0.908	45	5	0.66
Hedonism	385	7.205	1.243	18	2	0.42 *
Conformity	385	7.122	1.057	36	4	0.71
Security	385	6.947	0.946	45	5	0.60
Achievement	385	6.787	1.076	36	4	0.66
Universalism	385	6.609	1.140	72	8	0.81
Stimulation	385	6.607	1.363	27	3	0.76
Tradition	385	5.847	1.086	36	4	0.58
Power	385	5.010	1.419	36	4	0.81

M = Mean; SD = Standard deviation. MAX = Máximo; MIN = Mínimo; α = Reliability. * = The reliability of the Hedonism subscale was calculated by the correlation between its items as it did not have enough elements to do so by Crombach’s alpha.

**Table 2 ijerph-17-08315-t002:** Correlation between decision-making styles and value categories.

	Adaptive Style	Maladaptive Style
	Spearman’s *r*	Spearman’s *r*
Self-Direction	0.126 *	‒0.128 *
Benevolence	0.063	‒0.057
Conformity	0.086	‒0.086
Stimulation	0.110 *	‒0.045
Hedonism	0.035	‒0.039
Achievement	0.207 **	‒0.046
Power	0.140 **	0.071
Security	0.035	‒0.024
Tradition	0.089	‒0.003
Universalism	0.000	‒0.050

* *p* < 0.01. ** *p* < 0.05.

**Table 3 ijerph-17-08315-t003:** Correlation between decision-making styles and value categories based on gender.

Adaptive Style		Self-Direction	Benevolence	Conformity	Stimulation	Hedonism	Achievement	Power	Security	Tradition	Universalism
Male	Spearman’s *r*	0.113	−0.034	0.028	0.139	0.033	0.203 **	0.127	0.019	−0.004	−0.034
Female	Spearman’s *r*	0.146 *	0.158 *	0.182 *	0.023	0.004	0.205 **	0.121	0.059	0.196 **	0.087
**Maladaptive Style**		**Self-Direction**	**Benevolence**	**Conformity**	**Stimulation**	**Hedonism**	**Achievement**	**Power**	**Security**	**Tradition**	**Universalism**
Male	Spearman’s *r*	−0.202 **	−0.090	−0.065	−0.124	−0.071	−0.139	0.098	−0.014	−0.003	−0.073
Female	Spearman’s *r*	−0.062	−0.008	−0.087	0.048	0.017	0.028	0.056	0.030	0.043	0.024

* *p* < 0.01. ** *p* < 0.05.

**Table 4 ijerph-17-08315-t004:** Correlation between decision-making styles and value categories based on age.

Adaptive Style		Self-Direction	Benevolence	Conformity	Stimulation	Hedonism	Achievement	Power	Security	Tradition	Universalism
13-y.o.	Spearman’s *r*	0.230	0.160	0.290 *	0.185	0.003	0.229	0.073	0.105	0.191	0.143
14-y.o.	Spearman’s *r*	0.264 *	0.128	−0.075	0.314 *	0.268 *	0.323 *	0.419 **	0.236	0.150	0.067
15-y.o.	Spearman’s *r*	0.164	0.129	0.172	0.176	0.076	0.225	0.020	0.117	0.078	0.045
16-y.o.	Spearman’s *r*	0.024	−0.074	0.164	−0.058	−0.151	0.176	−0.031	−0.149	0.108	−0.086
17-y.o.	Spearman’s *r*	−0.028	0.077	−0.006	−0.010	0.026	0.084	0.081	−0.021	0.016	−0.124
18-y.o.	Spearman’s *r*	0.181	−0.022	0.112	0.025	0.071	0.431 **	0.531 **	−0.008	0.200	0.118
19-y.o.	Spearman’s *r*	0.239	−0.113	0.023	0.316	0.220	−0.011	−0.254	−0.039	−0.337	−0.221
**Maladaptive Style**		**Self-Direction**	**Benevolence**	**Conformity**	**Stimulation**	**Hedonism**	**Achievement**	**Power**	**Security**	**Tradition**	**Universalism**
13-y.o.	Spearman’s *r*	−0.072	0.015	−0.209	0.014	−0.042	0.019	0.130	−0.027	−0.016	−0.178
14-y.o.	Spearman’s *r*	−0.208	−0.025	−0.117	−0.244	0.043	−0.205	0.034	−0.114	−0.053	−0.063
15-y.o.	Spearman’s *r*	−0.085	−0.165	−0.124	0.023	−0.084	0.164	0.301 *	0.117	0.097	−0.010
16-y.o.	Spearman’s *r*	−0.125	0.042	−0.049	0.093	0.094	−0.112	−0.108	−0.038	−0.114	0.151
17-y.o.	Spearman’s *r*	−0.062	−0.002	−0.004	0.006	−0.121	−0.066	0.138	0.105	0.144	0.072
18-y.o.	Spearman’s *r*	−0.202	−0.108	0.071	−0.134	−0.166	−0.328 *	−0.298	0.007	−0.175	−0.128
19-y.o.	Spearman’s *r*	0.132	0.003	−0.381	0.104	0.093	0.269	0.579 *	0.052	0.373	0.179

* *p* < 0.01. ** *p* < 0.05.

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
