# Peer review of "Relationship Between Basic Human Values and Decision-Making Styles in Adolescents"

_ijerph, 2020, doi:10.3390/ijerph17228315_

Round 1
Reviewer 1 Report
The reviewed article on values is interesting. The issue of values is a fundamental variable in many studies on the psychosocial functioning of humans at different stages of life. The research undertaken by the authors should be highly valued in terms of potential reader interest. I do, however, have a few remarks which will perhaps make the study more readable:
- The introduction lacks a clear definition of 'universal values'. Of course, components have been identified according to the Schwartz typology, but there is a lack of a strict definition of the main variable;
- the research sample is very limited, which does not allow for generalisation. This element should be supplemented in the description of the research methodology;
- the title should be changed so as not to mislead readers that the study was carried out throughout Spain;
- other sociodemographic variables, such as residence, should be added to the description of the research sample;
- the tool is poorly described. I propose to add the psychometric properties of the tool (CFA) as well as internal consistency;
- the correlation coefficient used applies to variables with normal distribution. What was the distribution of variables for the collected results?
- In tables 2, 3, we do not have dependencies exceeding the average strength threshold. It is worth including this thread more strongly in the discussion section;
- for data analysis, it is worthwhile to apply a multi-factor analysis of variance and to show the prediction of individual indicators;
- It is also worthwhile to show the differences in the area of values in the context of other sociodemographic variables, such as place of residence;
- in the discussion we will find only six references to other studies - this is a little too little. The discussion needs to be deepened in the context of the hypotheses put forward;
- The text lacks the limitations of the research procedure and possible directions for further research.
I am keeping my fingers crossed for making amendments.
Author Response
Relationship between basic human values and decision-making styles in adolescents
First of all, we thank you for the suggestions you have made to our work as we are sure that they have served to improve the quality of the manuscript.
The changes made to the article are detailed below. In the sent text, the changes appear with change control to facilitate the revision of their location.
Introduction
- The title of the article has been modified for this other: Relationship between basic human values and decision-making styles in adolescents. Likewise, according to this change in the title, the abstract has been modified so readers are aware of the origin of the study participants.
- A clear definition of “universal values” has been included in the introduction to the article.
Sample
- The study sample was 2112 subjects, a resampling was made in which 385 participants were selected. This has been included in the methodology section. However, this procedure does not allow the generalization of the results, so this information has been included in the study limitations at the end of the manuscript.
- Information on the residence of the participants has been incorporated.
Instruments
- The psychometric properties of both scales fulfilled reliability and validity have been added.
Results pending
- After performing the normality tests using the Kolmogorov-Smirnof analysis, it was found that the probability of all the variables was less than .05, so the distribution of the scores did not fit the normal curve. For this reason, the correlations were carried out with Spearman's non-parametric test. The results were incorporated into the article.
- After analyzing which values in tables 2 and 3 did not give a statistically significant correlation, a tentative explanation was incorporated in the discussion section. Thus, both the significant and non-significant values Samples axiological profiles related to the behavioral development of each of them.
- The purpose of our study was the analysis of the relationships between basic social values and decision-making style. The analysis of variance proposed by the reviewer was performed but did not report significant results, so it was not incorporated into the final version of the article.
- The reviewer's proposal is very interesting, therefore we have taken it into account and it has been included as a possibility within the work as a possible future investigation.
Discussion
- More references have been included in the discussion according to the suggestion made by the reviewer. To do this, work from the last 5 years has been taken as a consultation criterion and thus guarantee the timeliness of the conclusions.
Conclusions
- The text of the conclusions has been reinforced and limitations have been included, as well as future research.
Without further ado, we thank you again for your appreciation.
Sincerely,

Reviewer 2 Report
- This is an interesting study.
- The manuscript contributes to furthering the analysis of basic human values and decision-making styles.
- The title, abstract, and research background are appropriate for the content of this text.
- The analysis is logical.
- The value of this study as a reference for the field of sociology and education is substantial.
- One suggestion is to add a description in the discussion section about how the findings of this study can be applied in the academic field.
Author Response
Relationship between basic human values and decision-making styles in adolescents
First of all, we thank you for the suggestions you have made to our work as we are sure that they have served to improve the quality of the manuscript.
The changes made to the article are detailed below. In the sent text, the changes appear with change control to facilitate the revision of their location.
Detailed changes
Discussion
- A proposal for the application of the conclusions has been included in the section of the same name. The objective has been to apply the results obtained from the research to daily practice.
Without further ado, we thank you again for your appreciation.
Sincerely,

Reviewer 3 Report
The paper explored the relationship between decision-making styles and values of Spanish adolescents. The analysis also examined age and gender on adaptive and maladaptive decision-making styles. The paper is valuable and interesting.
The introduction is effective. The concepts are described effectively. I also appreciate the thorough review of Schwartz Theory of Basic Human Values.
However, some suggestions are offered to guide its improvement:
Intro: It’s not clear what this study contributes to the field. This needs to be made clearer.
Literature review: Many articles are outdated, please add a couple more articles from 2019-2020.
Participants: what is the proportion of students by academic year?
Table 1: Can you include the alpha reliability value? And, the minimum and maximum?
Table 2 looks very odd. I suggest putting the adaptive/maladaptive style at the top and the values to the left, so that you would have fewer columns. Same with Table 3. Here is an example of what I mean: https://s3-us-west-2.amazonaws.com/courses-images/wp-content/uploads/sites/2714/2017/11/16174155/917f3163477d7cc5ade2024721104474.jpg
Italicize the r’s and the p’s in the results per APA style guidelines.
I am wondering why only basic analyses were conducted. Why was regression or logistic regression not used to further support/enhance the correlations? I suggest doing a sex comparison analysis such as a t-test or ANOVA to see if there are sex differences with the values.
The results also need to be contextualized using Schwartz Theory of Basic Human Values in the discussion section. How does this study extend prior work? What was the theoretical contribution?
As a whole, the study is well-designed. The findings are also highly valuable. I believe the study has much to offer.
Author Response
Relationship between basic human values and decision-making styles in adolescents
First of all, we thank you for the suggestions you have made to our work as we are sure that they have served to improve the quality of the manuscript.
The changes made to the article are detailed below. In the sent text, the changes appear with change control to facilitate the revision of their location.
Intruduction
- The information on the contribution of the study in the introduction of the manuscript has been reinforced.
- The number and variety of cited international documents published in the last 3 years has been increased, eliminating any reference prior to 2017 that did not provide substantial information to the study.
Sample
- The sociodemographic information on the distribution of students by academic year has been eliminated since this variable was not taken into account in the analyzes.
Instruments
- The alpha reliability value has been added for each scale, as well as the maximum and minimum.
Results pending
- Table 2 has been modified following the reviewer's guidelines.
- The r's and p's have been written in italics in the results following the recommendation of the reviewer.
- Regression analysis was also performed, but no significant results were reported.
Conclusions
- New interpretations of the results have been incorporated based on the proposals of Schwartz's theoretical model. To do this, taking into account the relationship between values and decision-making, it has been delved, mainly, in the behavioral development of said universal values.
Without further ado, we thank you again for your appreciation.
Sincerely,

Round 2
Reviewer 1 Report
Thank you for sending the text back for evaluation. The article has been significantly modified. All inaccuracies have been clarified and very precisely amended. The text now has a satisfactory level of readability. I would like to emphasise once again that the subject of basic human values is one of the basic variables for many studies, such as pedagogical, sociological and philosophical research. The study may therefore be helpful in other research. I congratulate the study and recommend the text for publication.
Reviewer 3 Report
Excellent revisions. I am satisfied with how the paper has shaped up to be. I believe this paper will make a positive contributions in the field.